# A New Method of Estimating Groundwater Evapotranspiration at Sub-Daily Scale Using Water Table Fluctuations

**Yonghong Su** [1,2,*], **Qi Feng** [1,2], **Gaofeng Zhu** [3], **Yunquan Wang** [4] **and Qi Zhang** [1,2]

1   Key Laboratory of Eco-Hydrology of Inland River Basin (CAS), Northwest Institute of Eco-Environment and Resources, CAS, Lanzhou 730000, China; qifeng@lzb.ac.cn (Q.F.); zhangqi21@nieer.ac.cn (Q.Z.)
2   College of Resources and Environment, University of Chinese Academy of Sciences, Beijing 100049, China
3   College of Earth and Environmental Sciences, Lanzhou University, Lanzhou 730000, China; zhugf@lzu.edu.cn
4   School of Environmental Studies, China University of Geosciences, Wuhan 430074, China; wangyq@cug.edu.cn
*   Correspondence: syh@lzb.ac.cn

**Abstract:** Riparian ecosystems fundamentally depend on groundwater, and accurate estimations of groundwater evapotranspiration ($ET_G$) are important for understanding ecosystem functionality and managing regional water resources. Over the past several decades, various methods have been proposed to estimate groundwater evapotranspiration based on water table fluctuations. However, the majority of methods cannot resolve sub-daily variations in $ET_G$. In this study, we proposed a new hydraulic theory-based $ET_G$ estimation method at a sub-daily time scale. To evaluate its performance, we employed a variety of measurements (i.e., water table levels, latent heat flux and soil water contents) at a riparian forest (*T. ramosissima*) in Northwest China from 25 July to 10 October in 2017. The results indicated that the proposed method can successfully estimate $ET_G$ at both sub-daily ($R^2 = 0.75$) and daily ($R^2 = 0.88$) time scales, but the variations in the specific yield under different water table conditions should be carefully taken into account. In addition, we investigated the seasonal variations in water uptake source of the riparian plant, and found that it had strong plasticity in water usage during the study period. That is, it consumed approximately equal amounts of soil water and groundwater when soil moisture was available, and tended to consume more groundwater for survival as the soil moisture was depleted. To verify the seasonal patterns of the water uptake of the riparian forest, systematic isotope-based studies are needed in future study.

**Keywords:** riparian forest; groundwater evapotranspiration; water table fluctuation; specific yield; water balance; plant water uptake

## 1. Introduction

Approximate 40% of the world's landmass is made up of drylands [1], where riparian ecosystems are very important in protecting biodiversity, maintaining ecological balance, and regulating ecosystem services [2]. Many riparian plants whose roots can tap perennial water tables depend on groundwater for their survival [3]. Therefore, accurate estimate of groundwater evapotranspiration ($ET_G$) is critical for understanding ecosystem functionality and managing regional water resources [4]. Up to now, various methods have been proposed to quantify $ET_G$ based on diurnal water table fluctuations [5–10]. Due to their cost-effectiveness and simplicity, these $ET_G$ estimation methods have been applied across diverse ecosystems [11–15].

Nevertheless, there are still some insufficiencies in the developments of the $ET_G$ estimation methods using water table fluctuations. First, the majority of methods assume that groundwater recovery rate is constant during a complete one-day period, and can only calculate $ET_G$ at a daily (or longer) time scale [5,6,9]. As first noted by Troxell [16] and evidenced by numerical simulations [7], the recovery rate is not constant over time, and must be estimated as a function of time. Until now, relatively little attention has been

paid to estimate the time-dependent groundwater recovery rate with the exception of that of Loheide [7] and Gribovszki et al. [8]. Second, the key parameter in the water table fluctuation methods, known as specific yield, is difficult to estimate, because neither in situ nor laboratory measurements capture its variability [4]. Previous studies have clearly documented that it not only depends upon the soil hydraulic properties and the water table depth, but is different under different water table conditions (i.e., declining, stable or rising) [17–21]. Unfortunately, systematic investigations of the variations in specific yield under different water table conditions are critically limited in present studies [11]. Finally, identifying the water source of plants is important for understanding the ecohydrological processes of riparian ecosystems [22,23]. Isotopic techniques have been widely used in previous studies [24–27]. However, sampling the isotopes of various water sources (i.e., soil water, groundwater and plant tissues) is technically challenging, destructive, expensive and time-consuming, which may hamper the assessment of the temporal variations in plant water source during growing seasons [28]. Thus, it is attractive to identify plant water sources by comprehensively using water table fluctuations and flux measurements [12].

Interest in ecohydrology and the need for finer resolution estimates of groundwater consumption increases are increasing. Here, we tried to develop a hydraulic theory-based method to estimate $ET_G$ at both sub-daily and daily scales by taking the time dependency of recovery rate into account. Based on 78-day continuous field observations including water table fluctuations, soil moisture and latent heat fluxes over a riparian ecosystem in Northwest China in 2017, the specific objectives of the present study were to: (1) develop a new hydraulic theory-based $ET_G$ estimation method at both sub-daily and daily scales; (2) investigate the variations in specific yield under different water table conditions, and evaluate its impacts on estimations of $ET_G$; and (3) identify the plant water uptake source during the whole study period.

## 2. Site Descriptions and Data Collection

The study site is located in the lower reaches of the Heihe River Basin, Gansu Province, China (101.1374° E, 42.0012° N; 873 m a. s. l.) (Figure 1). Situated in the hinterland of the Asian continent, the region has a continental climate with a mean annual precipitation of 42 mm and a mean annual potential evaporation of 2241 mm [29]. The soil texture profiles are silt loam with a clay interlayer. The riparian forest community is dominated by *T. ramosissima* with a mean height of 1.87 m and a density of 42 stem/hm$^2$ [30].

The field observation system was set up as part of the Heihe Watershed Allied Telemetry Experiment Research (HiWATER) project [31,32]. Latent heat fluxes were measured using the eddy covariance (EC) system at the height of 8.0 m, and post-processed by using EdiRe software (http://www.geos.ed.ac.uk/abs/research/micromet/EdiRe, accessed on 11 August 2021). Data gaps were filled using the look-up table method, and the imbalance of energy was corrected using the Bowen ratio method [33]. Continuous complementary measurements also included standard hydro-meteorological variables, such as rainfall, air temperature, relative humidity, wind speed/direction, net solar radiation, soil temperature and moisture at different depths (i.e., 0.02, 0.04, 0.1, 0.2, 0.4, 0.8, 1.2, 1.6 and 2.0 m). These data were logged every 30 min by a digital micro-logger [32–34] and are available from the National Tibetan Plateau/Third Pole Environment Data Center (http://data.tpdc.ac.cn, accessed on 11 August 2021).

In addition, one groundwater level monitoring well was installed near the EC flux tower. The depth of the well is about 4 m and the screen is located from the bottom of the well up to 1.0 m below surface using an 80 mm diameter PVC tubing. The groundwater level was measured using an automated pressure transducer (HOBO Water Level Logger-U20, Onset Computer Corp, Bourne, Massachusetts, USA) at half-hour intervals. The leaf area index (LAI) was not measured during the experiment period, and was extracted from MOD15A (https://modis.ornl.gov/, accessed on 11 August 2021) at 500 m spatial and 4 day temporal resolutions. An average of eight surrounding pixels around the EC tower were used to represent the seasonal variations in LAI.

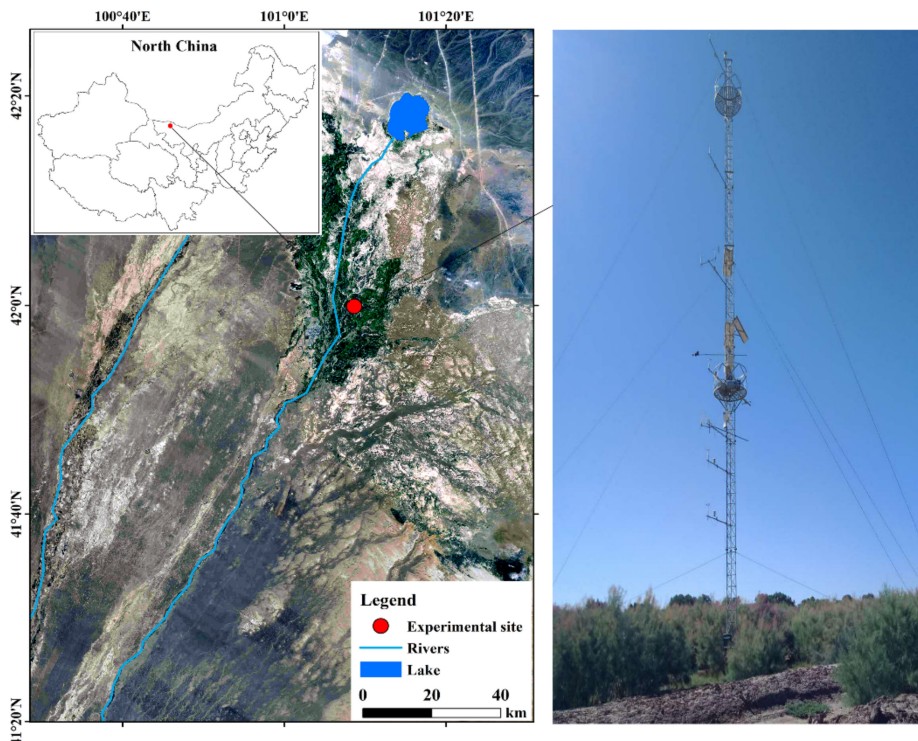

**Figure 1.** Experimental location and instrumentation setting at the Sidaoqiao station.

### 3. Methods

*3.1. Water Balance in the Riparian Zone*

Because of the scarce precipitation, surface evapotranspiration (ET; L T$^{-1}$) mainly comes from two water sources at our site: groundwater and soil water in the root layer. Therefore, total ET over the riparian zone at a daily scale could be estimated by using the water balance method:

$$ET(i) = ET_G(i) + ET_S(i) \tag{1}$$

where $i$ represents the day of the year; and $ET_G$ and $ET_S$ (all in units of (L T$^{-1}$)) the daily ET from groundwater and soil water, respectively. $ET_S$ can be calculated from the changes in soil water content in the root zone:

$$ET_S(i) = \sum_{l=1}^{n} d_l \times [\theta_l(i-1) - \theta_l(i)] \tag{2}$$

where $n$ is the number of the total soil layer; $d_l$ lL] the thickness of the $l$th soil layer ($l = 1, 2, \ldots, n$); $\theta_l(i)$ and $\theta_l(i-1)$ (all in L$^3$ L$^{-3}$) are the soil water content in the $l$th soil layer for the current day $i$ and previous day $i-1$, respectively. Inspired by previous studies i.e., [5–8,35], we here proposed a new hydraulic theory-based $ET_G$ estimation method using diurnal water table fluctuations.

*3.2. Hydraulic Theory-Based ET$_G$ Estimation Method*

At sub-daily scale, the change in water table in the riparian zone at time $t$ (hour) is controlled by the flow rate of groundwater from the background to the riparian zone, $q(t)$ [L T$^{-1}$], and the groundwater evapotranspiration rate, $ET_g(t)$ [L T$^{-1}$]:

$$S_y \frac{dWT}{dt} = q(t) - ET_g(t) \tag{3}$$

where $S_y$ (–) is the specific yield and WT [L] is the water table depth. Here, the lateral flow is defined as positive toward the riparian zone and the water table depth is defined as zero

at the land surface and becomes increasingly negative as the water table becomes deeper. Using Darcy's law and the Dupuit approximation, the groundwater flow rate toward the riparian zone can be formulated as [36]:

$$q(\text{WT}) = K_s \frac{H_0^2 - h^2}{2L} \propto a_1 \times \text{WT}^2 + a_2 \times \text{WT} + a_3 \tag{4}$$

where $K_s$ [L T$^{-1}$] is the saturated hydraulic conductivity of the aquifer system; $L$ [L] is the distance from the background to the riparian zone; $H_0$ [L] and $h$ [L] is the groundwater elevation in the background and the riparian zone, respectively (Figure 2); and $a_1$, $a_2$ and $a_3$ are hydrological coefficients (see details in Supporting Information), which are generally time-consuming and costly to measure directly [10]. We noticed that these coefficients can be derived from the water table records during times of zero ET$_g$. Combining Equations (3) and (4) by setting ET$_g$ = 0, we arrive at:

$$\frac{d\text{W}}{dt} = \frac{1}{S_y} \times q(\text{WT}) = \frac{1}{S_y} \times [a_1 \times \text{WT}^2 + a_2 \times \text{WT} + a_3] \tag{5}$$

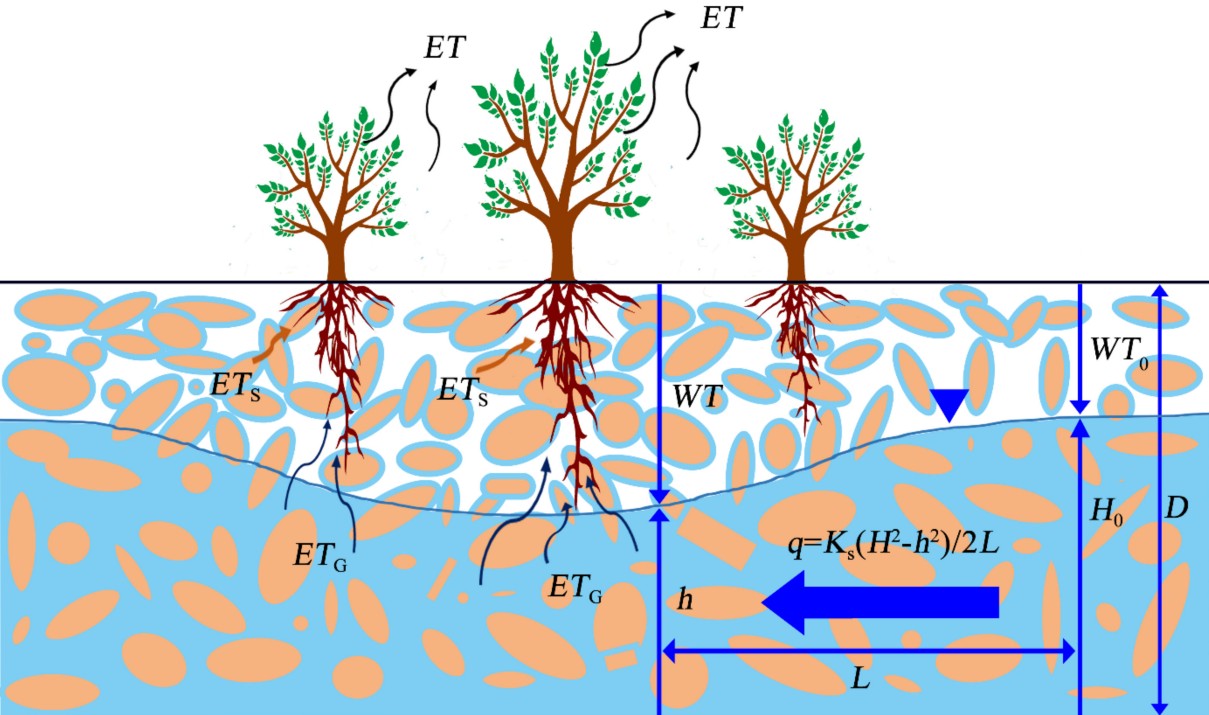

**Figure 2.** Conceptual diagram showing groundwater flow toward a riparian zone where it discharges via root uptake by phreato-phytic vegetation. *ET*s: ET from soil water [L T$^{-1}$]; *ET*$_G$: ET from groundwater [L T$^{-1}$]; WT$_0$ and WT: water table depth in the background and the riparian zone [L], respectively; $h_0$ and $h$: the groundwater elevation in the background and the riparian zone [L], respectively; $D$: the total depth of the aquifer [L]; $L$: the distance from the background to the riparian zone [L]; and $q$: the groundwater flow rate toward the riparian zone [L T$^{-1}$].

Thus, we established a relationship between the change in water table level over time $d\text{WT}/dt$ and water table level at sub-daily scale, and the coefficients can be easily determined by using the quadratic curve fitting method (Figure 3a). In practice, the change in water table level over time ($d\text{WT}/dt$) was calculated every 30 min using the slope estimated from the water table data at the time of the estimate, as well as the data before and after that time [7]. At our site, we found that observed ETs were close to zero during the time from midnight to 8 a.m. in the morning and from 7 p.m. to midnight on the day of interest. Thus, we propose using observations during these time intervals to estimate

the coefficients in Equation (5). Once these coefficients are determined, the groundwater flow rate $q(t)$ during the daytime period can be determined from the corresponding water table records using Equation (4) (Figure 3b). By rearranging Equation (3), the value of $ET_g$ at time $t$ can then be calculated as:

$$ET_g(t) = [a_1 \times WT^2(t) + a_2 WT(t) + a_3] - S_y \times \frac{dWT(t)}{dt} \qquad (6)$$

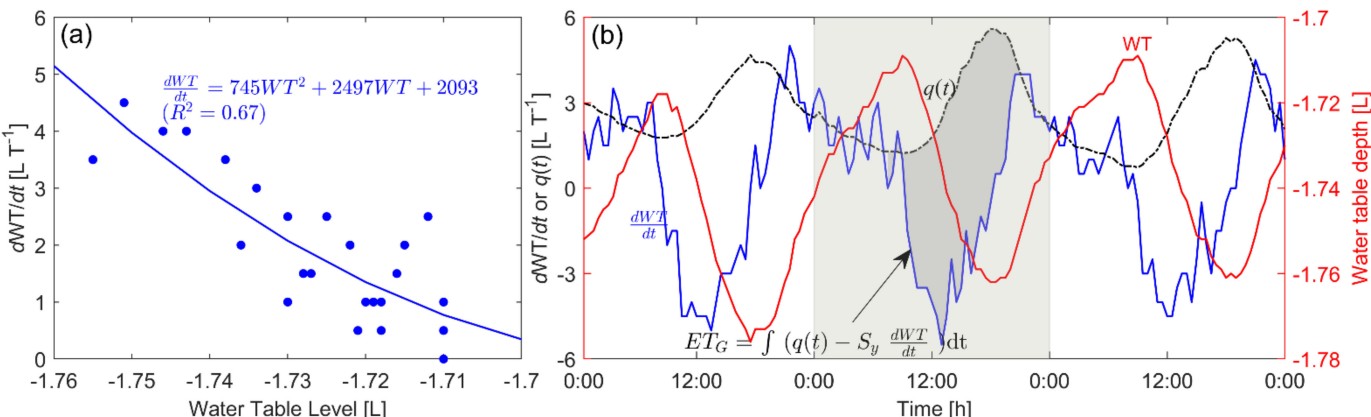

**Figure 3.** Illustration of the method proposed here for estimation of groundwater evapotranspiration based on water table fluctuation. (**a**) The rate of change in the water table dWT/dt as it changes with water table depth; the data show a quadratic relationship for the period of zero groundwater evapotranspiration; (**b**) diurnal variations in measured water table (WT) level (red line), calculated time-rate of change in measured water table level $dWT/dt$ (blue line), and the groundwater recovery rate $q(t)$ (dash black line) estimates at the study site for a 3-day sample period (10–12 August 2017). The day under investigation is highlighted in gray. Note that the method proposed here operates at the sub-daily time scale and has to be aggregated to obtain daily values $ET_G$ (dark gray).

Integrating Equation (6) over one-day interval $\Delta t$, we can obtain the accumulated daily $ET_G$:

$$ET_G = \int_{\Delta t} ET_g(t)dt = \int_{\Delta t} [a_1 \times WT^2(t) + a_2 WT(t) + a_3 - S_y \frac{dWT(t)}{dt}]dt \qquad (7)$$

where $ET_G$ [L T$^{-1}$] is daily ET from groundwater (Figure 3b). Thus, we can estimate groundwater evapotranspiration at both daily [$ET_G$; L T$^{-1}$] and sub-daily [$ET_g$; L T$^{-1}$] scales.

### 3.3. Determination of Specific Yield

The specific yield ($S_y$) is exceedingly difficult to estimate, and is highly variable in different water table conditions [4,5,11]. According to [12], we used the information available from the water balance method to estimate $S_y$. In brief, the value of $S_y$ was determined by minimizing the residual sum of squares (RSS) of Equation (1):

$$RSS = \sum_{i=1}^{T} [ET_{obs}(i) - (ET_S(i) + ET_G(i; \beta))]^2 \qquad (8)$$

where $T$ is the number of days during the study period; $ET_{obs}(i)$ is EC-measured evapotranspiration on the $i$th day ($i = 1, 2, \dots, T$); $ET_S(i) + ET_G(i; \beta)$ represents estimated evapotranspiration from soil water and groundwater on the $i$th day using the water balance method (Equation (1)); and $\beta$ is a coefficient that equals $S_y$ when RSS is minimized. In practice, we used a Monte Carlo (MC) algorithm to find the optimal value of $S_y$. The pseudo-code of the algorithm is given below:

Step 1: Calculating daily $ET_S$ from the soil moisture observations using Equation (2);

Step 2: Selecting a random value of $\beta$ from a prior interval of specific yield for silt loam soil (i.e., from 0 to 0.2), and calculating $ET_G$ using Equation (7) combined with selected $\beta$ and diurnal water table records;

Step 3: Calculating RSS using Equation (8);

Step 4: Repeating the Steps 2 and 3 for 10,000 times, and selecting the value of $\beta$ with minimum RSS as the optimal estimate of $S_y$. The best 500 values of $\beta$ with minimum RSSs were also used to calculate the standard deviation of $S_y$.

To account for the influences of water table conditions on $S_y$, we divided the entire observations into three different periods, and the optimal value of $S_y$ for each period was determined by using the MC algorithm. The three periods respectively represented a declining, constant and rising water table condition. Hereafter, we called this calibration procedure the 'dataset-by-dataset calibration'. In addition, we applied the MC algorithm on entire observations to obtain an optimal $S_y$ value for the whole study period, called the 'whole dataset calibration'. In this way, we can investigate the variations of $S_y$ under different water table conditions, and its impact on the performances of the proposed method throughout the study period.

### 3.4. Evaluating the Hydraulic Theory-Based $ET_G$ Estimation Method

After the values of $S_y$ were estimated, we used five statistical measures to evaluate the performance of the proposed method. These statistical measures were the coefficient of determination ($R^2$), slope, *y*-intercept, bias, root-mean square error (RMSE), and relative error (RE). The calculation formulas can be found in [37]. Among them, $R^2$ ranges between 0 and 1, with higher values indicating a good simulation result. Slope and *y*-intercept indicate how well the scatter plot between observed and estimated ET fits the 1:1 line. The values of bias, RMSE and RE illustrate the difference between observed and estimated ET, with lower values indicating a better model fit.

## 4. Results

### 4.1. Environmental Variables

Detailed information on the seasonality of key environmental and biological variables is essential to understand seasonal variation in plant water uptake. The seasonal change in daily net solar radiation ($R_n$; W m$^{-2}$), air temperature ($T_a$; °C), soil water content ($\theta$; m$^3$ m$^{-3}$), and leaf area index (LAI; m$^2$ m$^{-2}$) are illustrated in Figure 4. During the study period (form 25 July to 10 October), the daily mean $R_n$ varied from 33.1 to 208.7 W m$^{-2}$ with an average value of 124.8 W m$^{-2}$. The variation of mean daily $T_a$ has a similar trend to $R_n$, varying from 5.2 to 29.1 °C with an average value of around 20.1 °C (Figure 4a). The leaf area index (LAI; m$^2$ m$^{-2}$) showed a declining trend during the whole study period (Figure 4b). Interestingly, soil moisture ($\theta$; m$^3$ m$^{-3}$) only in the middle layers (80–160 cm) was observed to decrease gradually during the study period, and $\theta$ in other layers (0–80 cm and 160–200 cm) varied slightly (Figure 4c). The seasonal variation in water table can be divided into three periods (Figure 4d). From 25 July to 18 August, a generally declining trend of water level was observed. After that, the water table remained at a relatively constant level. In late September, the water table began to rise due to ceased plant water uptake.

### 4.2. Determination of Specific Yield

The values of specific yield ($S_y$) estimated using the minimum RSS method are shown in Figure 5. For the dataset-by-dataset calibration procedure, it was observed that $S_y$ exhibited significant variations under different water table conditions. That is, the maximum value (0.0359) was found during the water table declining period (25 July–18 August), followed by that (0.0206) during the water table stable period (19 August–22 September), and the minimum value (0.0137) occurred during the water table rising period (23 September–10 October). For the whole dataset calibration procedure, the value of $S_y$ was estimated to be

0.0251. This value was similar to that for the water table stable period (0.0206). This may be partly explained by the fact that the dataset during the water table stable period had a comparatively larger number of observations (i.e., 34 out of 77), and subsequently gained more weight in calculating the residual sum of squares.

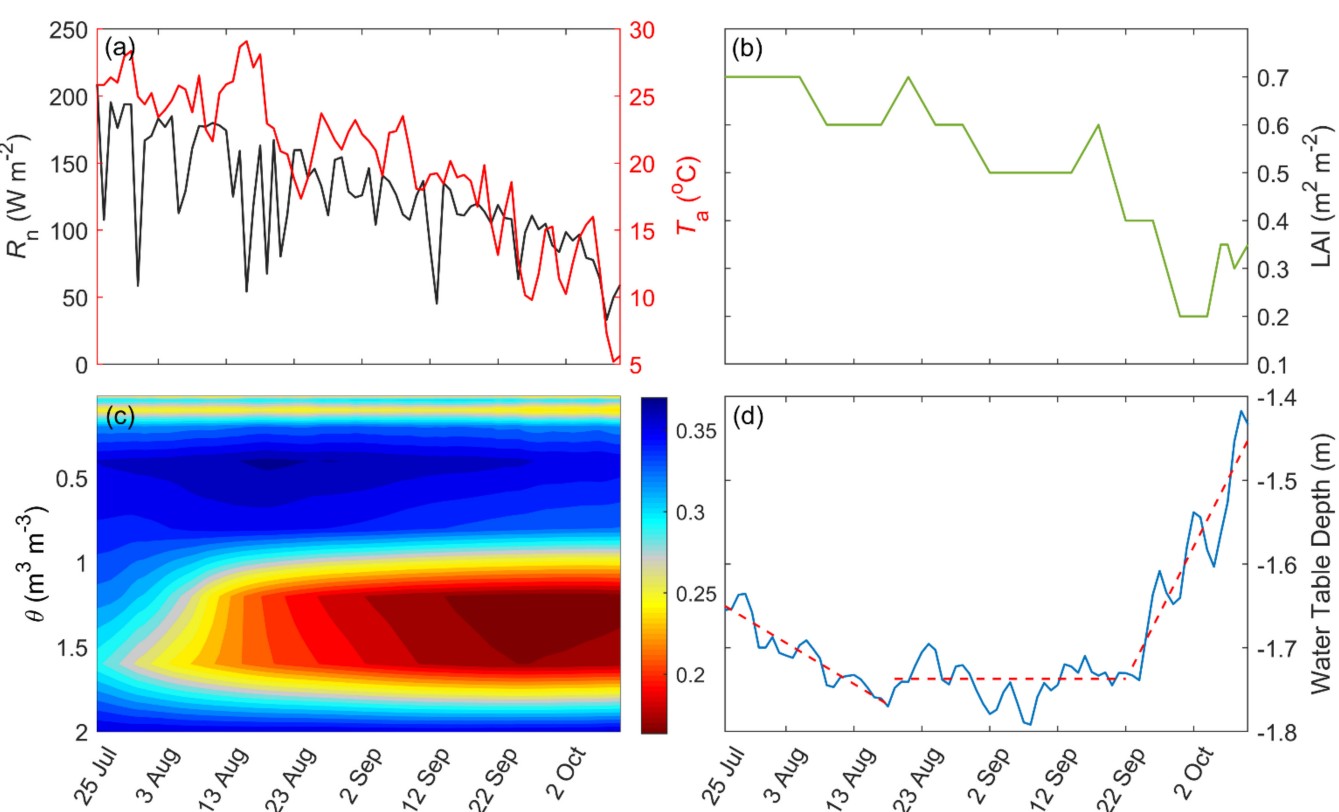

**Figure 4.** Seasonal variation in daily mean values of (**a**) $R_n$ (W m$^{-2}$) (black line) and $T_a$ (°C) (red line), (**b**) LAI (m$^2$ m$^{-2}$); (**c**) θ (m$^3$ m$^{-3}$) at 2, 20, 80, 120, 160 and 200 cm depth, and (**d**) water table depth (m) during the study period in the Sidaoqiao station.

### 4.3. Evaluating the Performances of the $ET_G$ Estimation Method

Having estimated $S_y$ as described above, we can obtain groundwater evapotranspiration at both a half-hourly ($ET_g$) and daily ($ET_G$) time step using Equations (6) and (7), respectively. To evaluate the performances of our proposed method, we compared the estimated daily ET using the water balance method with EC-measured values (Figure 6). The results indicated that the estimated daily ET using $S_y$ calibrated by different datasets agreed well with EC-measured values. The points in the plots of measured-versus-estimated daily ET fell tightly along the 1:1 line (slope = 0.91, intercept of 0.31 mm day$^{-1}$ and a correlation coefficient of 0.88), and the estimated daily ET fluctuated tightly with the measured values (bias= 0.06, RMSE = 0.85 and RE = 0.20) (Figure 6a,c). On the contrary, less satisfactory results were obtained by using $S_y$ calibrated by the whole dataset, with relatively low values of slope (0.66) and $R^2$ (0.76) and high RMSE (1.21) and RE (0.28) ((Figure 6b). In general, the estimated daily $ET_G$ using $S_y$ calibrated by the whole dataset was slightly underestimated and overestimated during the water table declining period and rising period (Figure 6c), respectively. This seems to be due to the significant differences in estimates of $S_y$ between the dataset-by-dataset and multi-dataset procedures during these two periods.

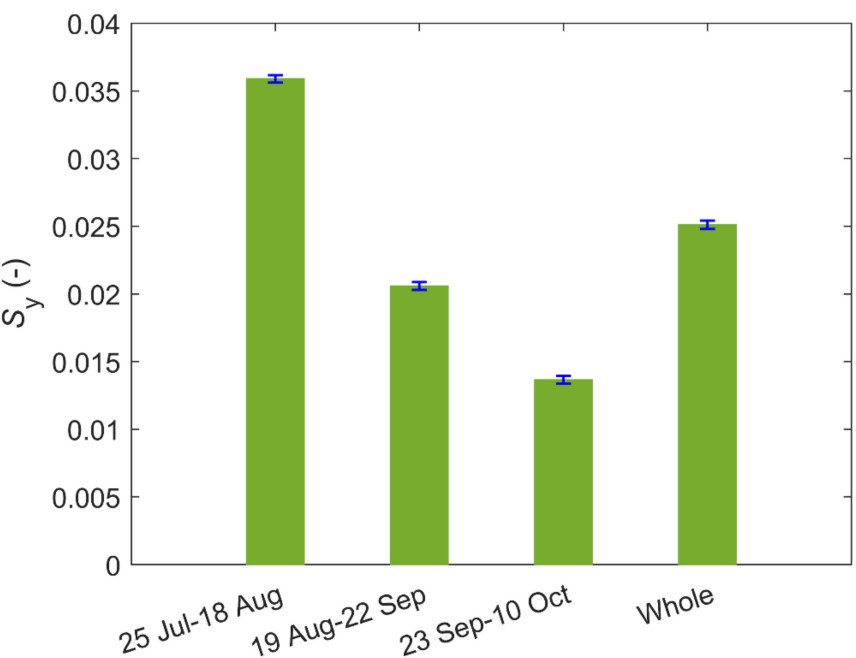

**Figure 5.** Specific yield ($S_y$) calculated by the water balance approach based on different datasets. The standard deviation was calculated using the best 500 values of Sy with minimum RSSs.

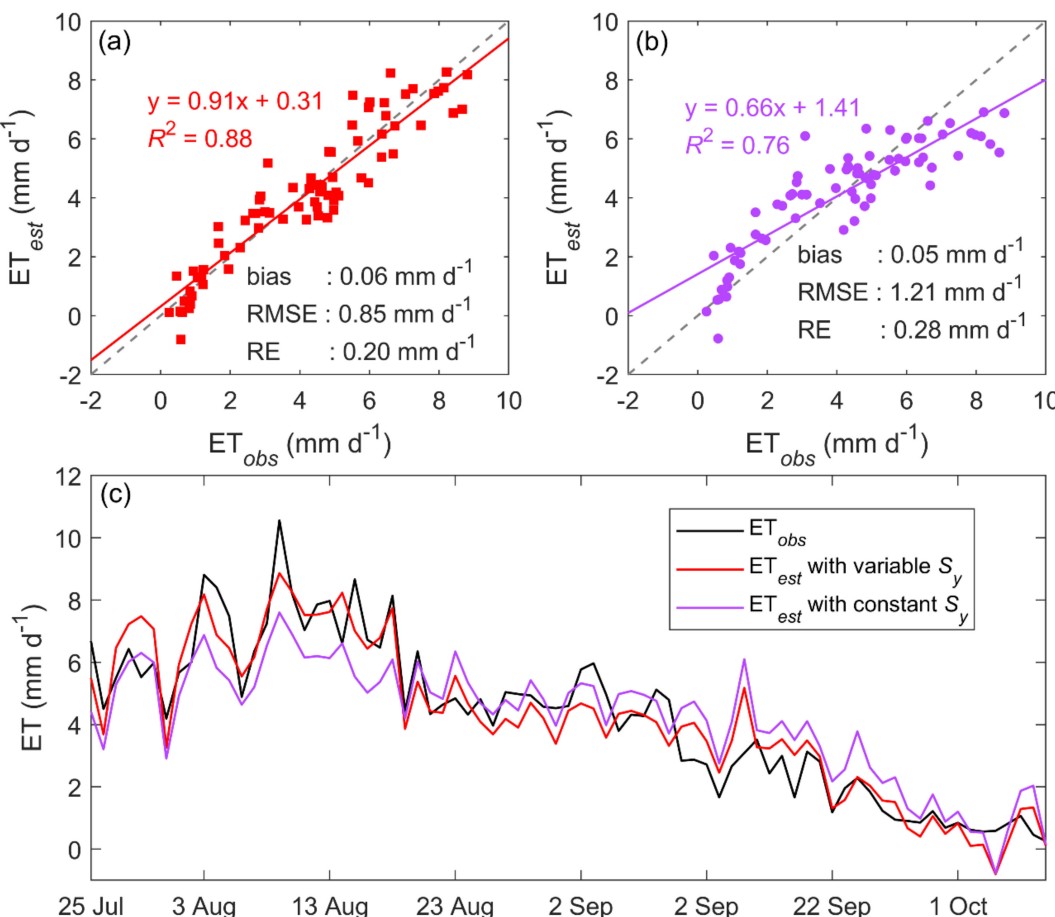

**Figure 6.** Comparisons between measured and estimated daily ET based on different Sy calibration procedures: (**a**) dataset-by-dataset procedure; (**b**) whole dataset procedure; and (**c**) seasonal variations of the measured and estimated daily ET based on different Sy calibration procedures.

Furthermore, we evaluated the performances of our proposed method at a half-hourly time step. Noticeably, the soil water evapotranspiration component in the water balance equation was not available at a half-hourly time step because the hour-to-hour variations in soil water content were generally not detectable. Thus, the slope of linear regression between EC-measured ET and estimated $ET_g$ at a half-hourly time step was expected to be less than 1 (Figure 7). Similar to the results observed at a daily time step, the estimated $ET_g$ using $S_y$ calibrated by the dataset-by-dataset procedure showed better correlations with measured ET than that using $S_y$ calibrated by the whole dataset (Figure 7a,b). In addition, it was observed that the diurnal pattern of estimated $ET_g$ and measured ET are very similar under different water conditions, despite that the magnitudes of $ET_g$ were different for the two $S_y$ calibration procedures (Figure 7c). Thus, it seemed that the method proposed here can successfully estimate groundwater evapotranspiration at different time steps, but the variations in $S_y$ should be carefully taken into account.

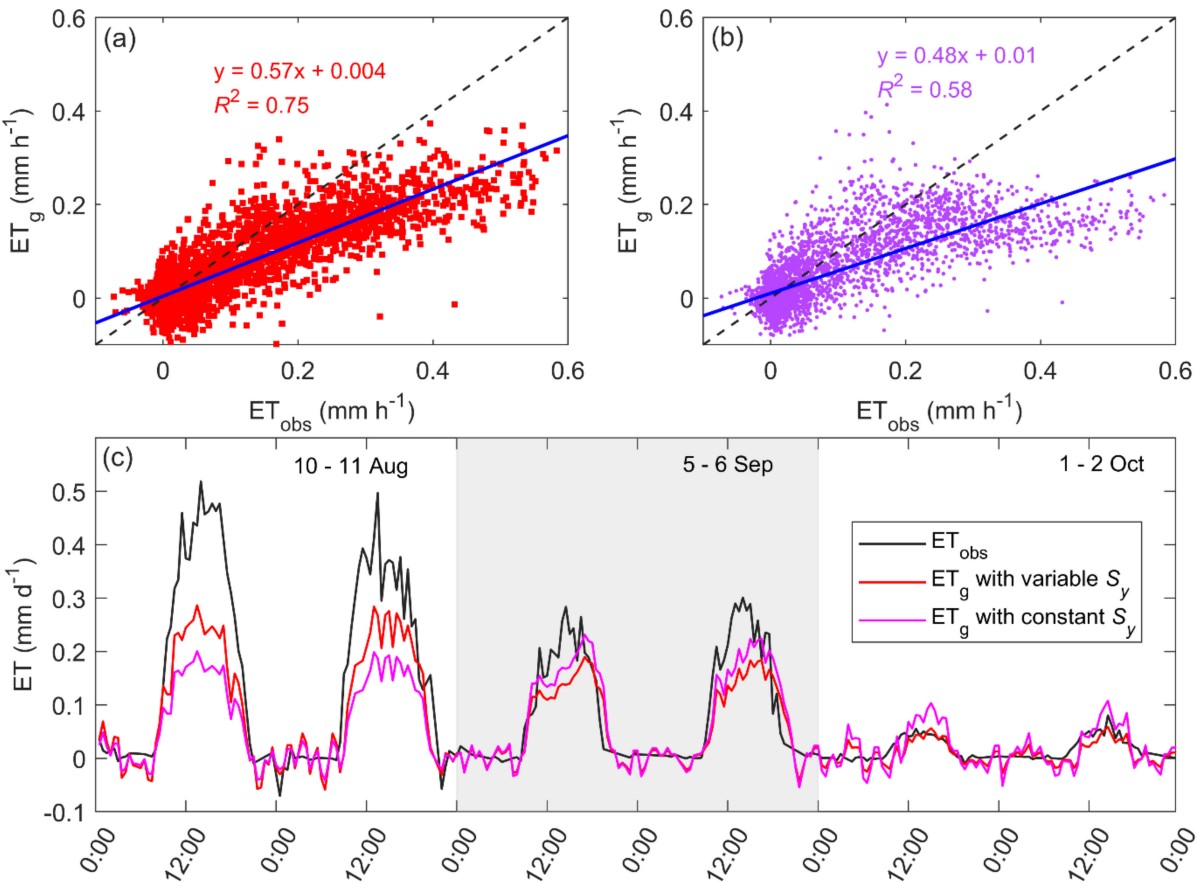

**Figure 7.** Comparisons between measured ET and estimated ETg at the sub-daily time scale based on different Sy calibration procedures: (**a**) dataset-by-dataset procedure; (**b**) whole dataset procedure; and (**c**) diurnal variations of the measured ET and estimated ETg based on different Sy calibration procedures.

### 4.4. Water Use Pattern

To reveal the water uptake pattern of the riparian plants, we plotted the seasonal variations in ET and its different components (i.e., $ET_S$ and $ET_G$ estimated using $S_y$ calibrated by different datasets) in Figure 8. The results indicated that during the whole study period the cumulative $ET_S$ and $ET_G$ were 121 mm and 202 mm (Figure 8b), respectively. The ratio of ET that comes from groundwater ($ET_G/ET$) and soil water ($ET_S/ET$) was 0.62 and 0.38, respectively. Thus, groundwater was the main source for plant water consumption during the whole study period. Interestingly, we observed that the riparian plant had strong

plasticity in water uptake during the whole study period. When soil moisture is available (i.e., from 25 July to 12 August), the plant consumed approximately equal amounts of soil water and groundwater resources (Figure 8a), and the ratios of $ET_G/ET$ and $ET_S/ET$ were 0.54 and 0.46 (Figure 8b), respectively. As the soil moisture was depleted (i.e., after 13 August), the plant tended to consume more groundwater for survival (Figure 8a), and the ratios of $ET_G/ET$ (>0.64) were larger than that of $ET_S/ET$ (<0.36) (Figure 8b).

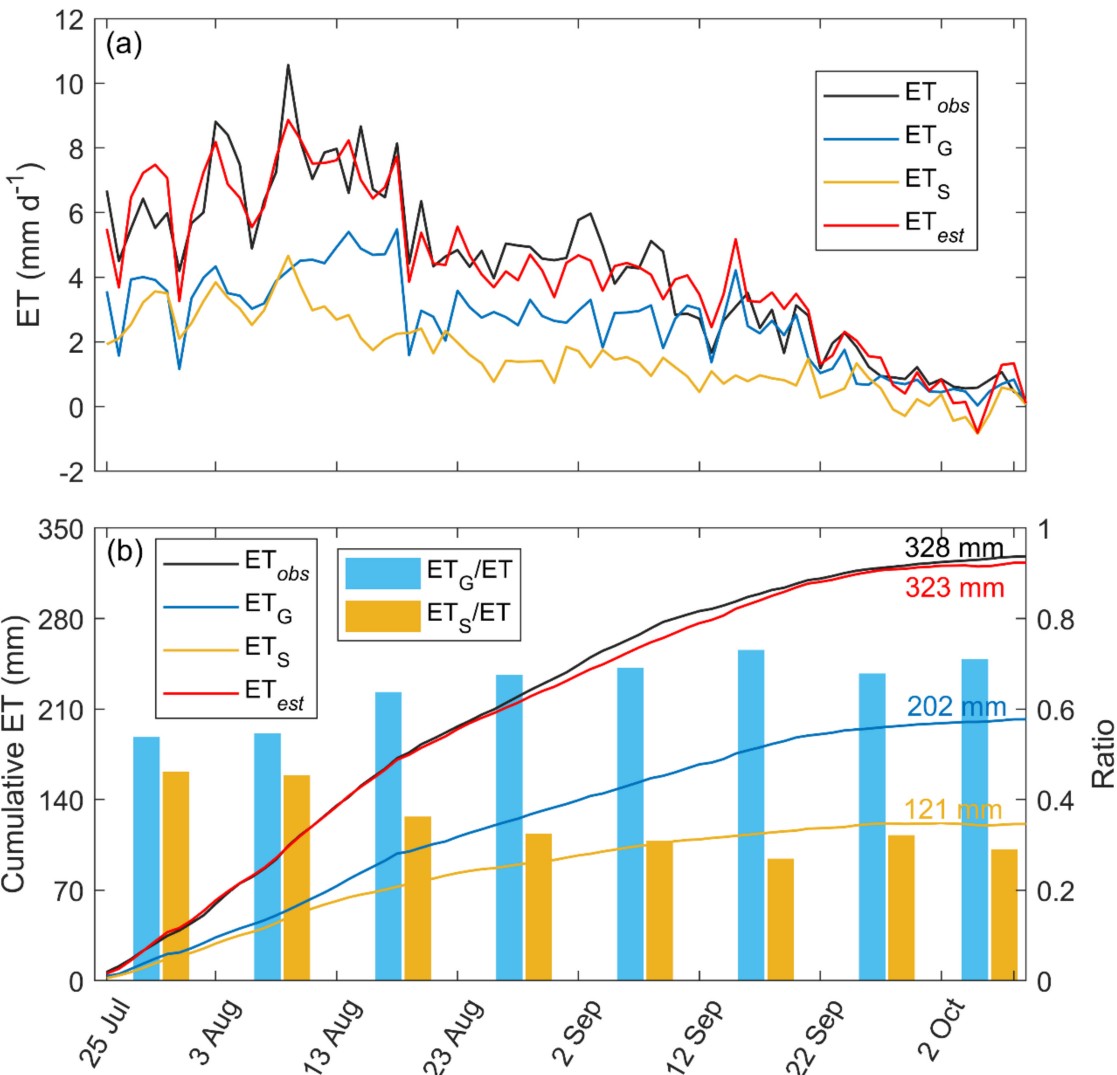

**Figure 8.** Seasonal variations in (**a**) measured and estimated daily ET, as well as the different components of ET (i.e., ETS and ETG); (**b**) cumulative ET and its different components (left axis) and the ratios of ETG/ET and ETs/ET at 10 day intervals (right axis) during the whole study period.

## 5. Discussion

The specific yield ($S_y$) is of crucial importance for estimating $ET_G$ based on water table fluctuations because its error is translated directly to the final estimates [5,10,38]. In this study, the values of $S_y$ estimated using the minimum RSS method fell within the ranges of the literature reports for silt loam soil. For example, Johnson [39] reported that $S_y$ for silt loam varied from 0.01 to 0.2. Singhal and Gupta [40] documented that the values of $S_y$ for rocky silt loam ranged from 0.02 to 0.05. In addition, we also calculated the $S_y$ values using the method proposed by Loheide et al. [21], which ranged from 0.0297 to 0.0353 with a mean of 0.0339 during the whole study period (see Supporting Information Figure S1). Thus, we thought that the minimum RSS method was suitable to obtain proper $S_y$ values

for $ET_G$ estimates in our site. Interestingly, we found that $S_y$ varied significantly under different water table conditions, and its value for a declining water table condition was about three times that of a rising water table condition. This can be explained by the fact that encapsulated air in the aquifer, which can be as high as 20% of soil porosity [41], reduces the value of $S_y$ during the water table rising periods. Therefore, it is important to take the variations in $S_y$ under different water table conditions into account for long-term groundwater evapotranspiration estimations.

We further compared the performance of our proposed method with the methods developed by Loheide [7] and Griovszki et al. [8]. The Gribovszki method uses the rate of the water table change $d\mathrm{WT}/dt$ [L T$^{-1}$] to estimate the groundwater recharge rate $q(t)$. For each day, and maximum and minimum $q(t)$ were obtained by selecting the largest positive rate of $d\mathrm{WT}/dt$ and the mean of $d\mathrm{WT}/dt$ between midnight and 6 a.m., respectively; then it is assumed that $q(t)$ behaves linearly between two consecutive estimations. Loheide [7] proposed to estimate $q(t)$ as a function of the detrended water table. The function, set up using data between midnight to 6 a.m. of two subsequent days, is assumed to be approximately linear during each day. Overall, the performance of our method was very similar to the Gribovszki method (Figure 9). However, we found that during the nighttime period (i.e., from 9 p.m. of the previous night to 6 a.m. in the morning) the estimated $q(t)$ by the Gribovszki method was generally higher than the rate of $d\mathrm{WT}/dt$ (Figure 9a). Thus, unreasonable positive $ET_g$ was obtained by this method during the nighttime (Figure 9b). In addition, the maximum $q(t)$ estimated by our method was generally higher and occurred earlier than that estimated by the Gribovszki method. Thus, the values of $ET_g$ estimated by our method were slightly higher than that of the Gribovszki method in the afternoon. On the contrary, the method proposed by Loheide [7] performed relatively unsatisfactorily. This may be mainly because the assumptions of the Loheide method (i.e., the head at the recovery source is constant or follows the general trend of the water table) may not be met at our site. Fahle and Dietrich [10] have systematically compared the performances of six widely used groundwater evapotranspiration estimation methods, and also found that the Loheide method performed considerably worse than other methods.

The results of our study revealed that the riparian plant exhibit substantial water uptake plasticity during different growing stages. When soil moisture was adequate, the riparian plant mainly used soil water. As soil water depleted, it then tended to depend on groundwater (Figure 8). Similar water use strategies were also observed in other arid and semi-arid plant communities [24–27,42–44]. Furthermore, the estimated ratios of $ET_G/ET$ by our method agreed with the results using the stable isotopic methods. For example, Wu et al. [43] documented that the ratios of $ET_G/ET$ of *T. ramosissima* at the southern edge of the Gurbantonggut desert were more than 0.43 in summer and autumn. Zhao et al. [45] reported that the ratio of plant water uptake that from groundwater of *T. ramosissima* in the lower reaches of Heihe River Basin was 0.49 in wet seasons and increased to 0.90 in dry seasons. To assess the seasonal patterns of water uptake of the riparian plant, a systematic collection of stable isotopes of different water sources (i.e., soil water, groundwater and xylem water) is needed in the future studies.

Finally, it should be noticed that there are still some uncertainties in our method. First, we used information from the water balance method to help estimate the values of $S_y$. In this study, the profile of soil moisture was measured only at one site, and it may be difficult to capture the spatial variability in soil moisture. Thus, there may be considerable uncertainty in the soil moisture portion of the water balance equation. To overcome this problem, a soil moisture monitoring network is planned to be set up at our site. Second, the periods from midnight to 8 a.m. in the morning and from 7 p.m. to midnight on the day of interest were selected for the recovery analysis at our site. However, this is a somewhat subjective choice of time period. To apply this method at other sites, a slightly different time period may be better for calibrating the parameters in Equation (5). Finally, if hydraulic redistribution is a significant process, plant water uptake from groundwater during the

night is not zero. Thus, the fundamental assumption of diurnal water table fluctuation based methods will be violated [46].

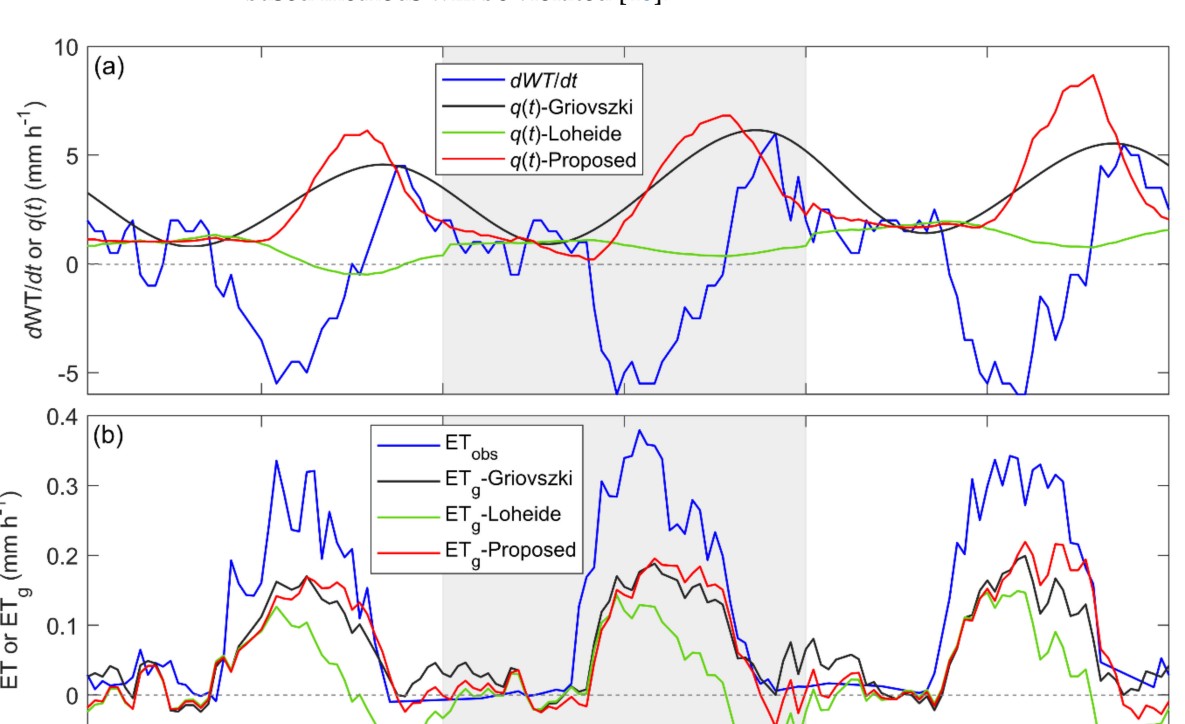

**Figure 9.** Illustrations of the performances of different sub-daily groundwater evapotranspiration estimation methods for a 3-day sample period (1–3 September 2017). (**a**) diurnal variations in measure water table level, calculated time-rate of change in measured water table level, and the groundwater recovery rate $q(t)$ estimated by the 3 different methods; (**b**) diurnal variations of the measured ET and estimated ETg by the 3 different methods.

## 6. Conclusions

In this study, we developed a new hydraulic theory-based method to estimate groundwater evapotranspiration at a sub-daily time scale. The diurnal pattern of estimated groundwater evapotranspiration exhibited similar trends to EC-measured evapotranspiration. Thus, it seemed that this method permits improvement in the estimation of groundwater evapotranspiration just by using diurnal water table fluctuations. Another important problem in applying water table fluctuation methods is to properly determine the values of specific yield ($S_y$). Our study revealed that the $S_y$ values determined by the minimum residual sum of squares method were consistent with those reported in the literature for similar soil, and varied considerably under different water table conditions. For long-term groundwater evapotranspiration estimations, the variations of $S_y$ under different water table conditions (i.e., falling, stable and rising) should be properly taken into account. In the future, stable isotope-based studies are needed to evaluate the accuracy of the method in estimating groundwater evapotranspiration rates.

**Supplementary Materials:** The following supporting information can be downloaded at: https://www.mdpi.com/article/10.3390/w14060876/s1, Figure S1: Comparisons of estimated Sy values by different methods.

**Author Contributions:** Conceptualization, Y.S. and Q.F.; software, G.Z.; data curation, Q.Z.; writing—review and editing, Y.W. All authors have read and agreed to the published version of the manuscript.

**Funding:** This research was founded by the National Natural Science Foundation of China (Nos. 42071138 and 42171019), and by the Natural Science Foundation of Gansu Province (21JR7RA054).

**Data Availability Statement:** The MOD15A products used in this study are available at https: //modis.ornl.gov/. The flux data are available from the National Tibetan Plateau/Third Pole Environment Data Center (http://data.tpdc.ac.cn).

**Acknowledgments:** We would like to thank Helen Jing for her continuous help during the review process and careful edits on the draft of the manuscript. We also thank the two anonymous reviewers for their critical reviews and helpful comments.

**Conflicts of Interest:** The authors declare no conflict of interest.

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
