# Peer review of "A New Method of Estimating Groundwater Evapotranspiration at Sub-Daily Scale Using Water Table Fluctuations"

_water, doi:10.3390/w14060876_

Round 1

Reviewer 1 Report

The authors propose a new methodology in assessing evapotransporation in aquifers at sub-daily level considering water table conditions. The paper is well written, metholody is clearly exposed and results are deeply analysed, Furthermore, paper makes clear their limitations and future research. I only recommend minor revisions in several parts of the document, listed below.

Lines 17-18: provide results, correlations, statistics

Lines 19: "... should be taken into account carefully..." Why? Briefly explain.

Line 77: provide location map.

Lines 91-92: explain the methods (include in Appendix)

Line 99: longitude and latitute--> revise superscripts and subscripts in several lines.

Lines 86-97: consider move to methodology section.

Figure 2b: provide legend

Figure 4b: add legend

References do not follow rules of journal 

Author Response

We have reply the reviewer's comments. See details in the file named 'Responses to Reviewers'.

Reviewer 2 Report

See Attached

Author Response

We have relay the comments from the reviewer. 
